# Neonatal outcome in 29 pregnant women with COVID-19: A retrospective study in Wuhan, China

Yan-Ting Wu[1]⊗, Jun Liu[2,3]⊗, Jing-Jing Xu[1]⊗, Yan-Fen Chen[4]⊗, Wen Yang[3], Yang Chen[3], Cheng Li[1], Yu Wang[1], Han Liu[1], Chen Zhang[1], Ling Jiang[1], Zhao-Xia Qian[1], Andrew Kawai[5], Ben Willem Mol[5], Cindy-Lee Dennis[6], Guo-Ping Xiong[4]*, Bi-Heng Cheng[2]*, Jing Yang[2]*, He-Feng Huang[1]*

1 International Peace Maternity and Child Health Hospital, School of Medicine, Shanghai Jiao Tong University, Shanghai, China, 2 Renmin Hospital, Wuhan University, Wuhan, China, 3 Wuhan Children's Hospital (Wuhan Maternal and Child Healthcare Hospital), Tongji Medical College, Huazhong University of Science and Technology, Wuhan, China, 4 The Central Hospital of Wuhan, Tongji Medical College, Huazhong University of Science and Technology, Wuhan, China, 5 Department of Obstetrics and Gynaecology, Monash University, Clayton, Victoria, Australia, 6 Bloomberg Faculty of Nursing, University of Toronto, Ontario, Canada

⊗ These authors contributed equally to this work.
* xionggp_whszxyy@163.com (GPX); Biheng_Cheng@whu.edu.cn (BHC); 13507182023@163.com (JY); huanghefg@sjtu.edu.cn (HFH)

**Data Availability Statement:** All the research data are available at the ResMen Manager of Chinese Clinical Trial Registry (www.medresman.org), and the registration number is ChiCTR2000031954

## Abstract

### Background

As of June 1, 2020, coronavirus disease 2019 (COVID-19) has caused more than 6,000,000 infected persons and 360,000 deaths globally. Previous studies revealed pregnant women with COVID-19 had similar clinical manifestations to nonpregnant women. However, little is known about the outcome of neonates born to infected women.

### Methods and findings

In this retrospective study, we studied 29 pregnant women with COVID-19 infection delivered in 2 designated general hospitals in Wuhan, China between January 30 and March 10, 2020, and 30 neonates (1 set of twins). Maternal demographic characteristics, delivery course, symptoms, and laboratory tests from hospital records were extracted. Neonates were hospitalized if they had symptoms (5 cases) or their guardians agreed to a hospitalized quarantine (13 cases), whereas symptom-free neonates also could be discharged after birth and followed up through telephone (12 cases). For hospitalized neonates, laboratory test results and chest X-ray or computed tomography (CT) were extracted from hospital records. The presence of antibody of SARS-CoV-2 was assessed in the serum of 4 neonates.

Among 29 pregnant COVID-19-infected women (13 confirmed and 16 clinical diagnosed), the majority had higher education (56.6%), half were employed (51.7%), and their mean age was 29 years. Fourteen women experienced mild symptoms including fever (8), cough (9), shortness of breath (3), diarrhea (2), vomiting (1), and 15 were symptom-free.

(http://www.medresman.org.cn/pub/cn/proj/projectshshow.aspx?proj=1810).

**Funding:** YTW: National Key Research and Development Program of China (2018YFC1002804), http://www.most.gov.cn; YTW: National Key Research and Development Program of China (2016YFC1000203), http://www.most.gov.cn; CL: COVID-19 Prevention and Control Program of International Peace Maternity and Child Health Hospital, School of Medicine, Shanghai Jiao Tong University (2020-COVID-19-04), https://www.ipmch.com.cn. The funders had no role in study design, data collection and analysis, decision to publish, or preparation of the manuscript.

**Competing interests:** I have read the journal's policy and the authors of this manuscript have the following competing interests: BWM is supported by a NHMRC Investigator grant (GNT1176437). BWM reports consultancy for ObsEva, Merck KGaA, iGenomix and Guerbet. These consultancies are not related to the current work. All other authors report no potential competing interests.

**Abbreviations:** AGA, appropriate for gestational age; ALB, albumin; ALT, alanine aminotransferase; ASD, atrial septal defect; AST, aspartate transaminase; BMI, body mass index; BUN, urea nitrogen; CK, creatine kinase; COVID-19, coronavirus disease 2019; CREA, creatinine; CRP, C-reactive protein; CS, cesarean section; CT, computed tomography; GDM, gestational diabetes mellitus; GGO, ground-glass opacity; HBV, hepatitis B virus; HIE, hypoxic-ischemic encephalopathy; ICU, intensive care unit; IgG, immunoglobulin G; IgM, immunoglobulin M; IQR, interquartile range; LDH, lactate dehydrogenase; LGA, large for gestational age; LYM%, lymphocyte percentage; LYM, lymphocyte count; NEC, necrotizing enterocolitis; NEU%, neutrophil percentage; NEU, neutrophil count; NICU, neonatal intensive care unit; NRDS, neonatal respiratory distress syndrome; PCT, procalcitonin; PDA, patent ductus arteriosus; PHEIC, Public Health Emergency of International Concern; PLT, platelet count; qRT-PCR, quantitative real time-PCR; SARS, severe acute respiratory syndrome; SARS-CoV-2, severe acute respiratory syndrome coronavirus 2; SD, standard deviation; TP, total protein; UA, uric acid; WBC, white blood cell count.

Eleven of 29 women had pregnancy complications, and 27 elected to have a cesarean section delivery.

Of 30 neonates, 18 were admitted to Wuhan Children's Hospital for quarantine and care, whereas the other 12 neonates discharged after birth without any symptoms and had normal follow-up. Five hospitalized neonates were diagnosed as COVID-19 infection (2 confirmed and 3 suspected). In addition, 12 of 13 other hospitalized neonates presented with radiological features for pneumonia through X-ray or CT screening, 1 with occasional cough and the others without associated symptoms. SARS-CoV-2 specific serum immunoglobulin M (IgM) and immunoglobulin G (IgG) were measured in 4 neonates and 2 were positive. The limited sample size limited statistical comparison between groups.

## Conclusions

In this study, we observed COVID-19 or radiological features of pneumonia in some, but not all, neonates born to women with COVID-19 infection. These findings suggest that intrauterine or intrapartum transmission is possible and warrants clinical caution and further investigation.

## Trial registration

Chinese Clinical Trial Registry, ChiCTR2000031954 (Maternal and Perinatal Outcomes of Women with coronavirus disease 2019 (COVID-19): a multicenter retrospective cohort study).

## Author summary

### Why was this study done?

- Previous studies suggest COVID-19 infections in pregnant women are not more severe than in women of similar age.
- However, little is known about the outcome of neonates born to COVID-19-infected women.

### What did the researchers do and find?

- We analyzed the clinical features of all pregnant women with COVID-19 infection delivered in 2 of the 5 designated general hospitals in Wuhan, China between January 30 and March 10, 2020, and their neonates.
- Among the 30 neonates born to 29 pregnant COVID-19-infected women (1 set of twins), 5 were diagnosed as having COVID-19 (2 confirmed and 3 suspected), and the other 12 neonates presented radiological features of pneumonia, but none were diagnosed with pneumonia.
- Three neonates developed necrotizing enterocolitis, although none of them were premature.

## What do these findings mean?

- A considerable number of, but not all, neonates born to COVID-19-infected women developed COVID-19 (5/30) or had pneumonia-like radiological features (17/30).

- Intrapartum precaution should not be underestimated, and intrauterine transmission should be considered based on current evidence.

- Term neonates born to COVID-19-infected women are vulnerable to develop necrotizing enterocolitis.

## Introduction

Since December 2019, a novel coronavirus called severe acute respiratory syndrome coronavirus 2 (SARS-CoV-2) has spread through human-to-human transmission across China [1] and now internationally to over 160 countries. The World Health Organization initially classified the outbreak of the coronavirus disease 2019 (COVID-19) as a Public Health Emergency of International Concern (PHEIC), and on March 11, 2020, they upgraded it to a pandemic [2]. According to the Chinese Center for Disease Control and Prevention (China CDC), 25.1% of the 44,672 confirmed cases in mainland China were women of reproductive age [3,4]. Pregnant women experience immunologic and physiologic changes, which make them potentially more susceptible to viral respiratory infections, including respiratory syncytial virus, influenza virus, and SARS-CoV [5]. There is growing evidence that COVID-19 infections in pregnant women are not more severe than in age-matched women where symptoms are typically mild [6–12], and childbirth did not aggravate the course of the illness or chest computed tomography (CT) features of COVID-19 [13]. In addition, previous research suggests infants under 1 year old are susceptible to COVID-19 when in contact with infected family members [14]. However, for neonates born to infected women, little is known about their susceptibility.

In this retrospective study, we analyzed maternal and neonatal clinical characteristics of all women with confirmed or clinical diagnosed COVID-19 infection who gave birth in 2 of the 5 general hospitals in Wuhan, China and followed up their neonates through telephone interview or collecting their clinical data in the only designated children's hospital to assess neonatal outcomes associated with maternal COVID-19 infection.

## Methods

### Study design and participants

All pregnant women diagnosed with COVID-19 who gave birth between January 30 to March 10, 2020 in 2 designated general hospitals in Wuhan, China (Renmin Hospital, Wuhan University, and Central Hospital of Wuhan, Tongji Medical College, Huazhong University of Science and Technology) were included in this retrospective study. Clinical characteristics, examination results, and treatment course were extracted from their medical records.

All procedures performed in this study involving human participants were in accordance with the ethical standards of the Medical Ethical Committee of Wuhan Children's Hospital (Wuhan Maternal and Child Healthcare Hospital), Tongji Medical College, Huazhong University of Science and Technology (WHCH2020014), Central Hospital of Wuhan, Tongji Medical College, Huazhong University of Science and Technology (2020–28), and Renmin Hospital,

Wuhan University (WDRY2020-K097). For this retrospective study in clinical practice, the requirement for written informed consent was waived by the ethics committee.

## Maternal data

We followed the diagnostic criteria for COVID-19 in pregnant women according to the New Coronavirus Pneumonia Prevention and Control Program (fifth, sixth, and seventh edition) [15–17] published by the National Health Commission of China (specific diagnostic criteria were shown in S1 Text). In women where a PCR-kit for SARS-CoV-2 RNA was not available, the diagnosis was made with chest CT scan.

For mothers, we collected sociodemographic data (maternal age, body mass index [BMI], educational attainment [>high school or not], and occupation [employed specifically at a hospital, employed but not at a hospital, or unemployed]), maternal parity (primiparous or multiparous), medical history records (fever, cough, shortness of breath, diarrhea, and vomiting), laboratory tests, chest CT images, throat swab tests, use of mechanical ventilation, and intensive care unit [ICU] admission. Laboratory tests included routine antenatal blood tests (white blood cell count [WBC], lymphocyte count [LYM], lymphocyte percentage [LYM%]), levels of C-reactive protein (CRP) in serum, biochemical indicators of hepatic and renal function (aspartate transaminase [AST], alanine aminotransferase [ALT], creatine kinase [CK], lactate dehydrogenase [LDH], total protein [TP], albumin [ALB], uric acid [UA], creatinine [CREA], and urea nitrogen [BUN]), and postnatal blood routine tests and levels of CRP. For all women, we registered pregnancy outcomes, i.e., gestational age at delivery, mode of delivery, and occurrence of gestational hypertensive disorder, gestational diabetes mellitus, premature rupture of membranes, and fetal distress.

## Neonatal data

Neonates who need hospital care or who were admitted on parental request for quarantined observation were admitted to a designated children's hospital (Wuhan Children's Hospital [Wuhan Maternal and Child Healthcare Hospital], Tongji Medical College, Huazhong University of Science and Technology). When parents did not opt for quarantined observation, neonates were discharged after birth. For these neonates, the outcome was obtained through parental telephone interview.

Neonatal data collected included sex, birthweight, 1- and 5-minute Apgar scores, congenital anomalies, fever, respiratory distress and neonatal ICU (NICU) admission. Neonates admitted at NICU care had additional data collected including throat swabs, anal swabs, as well as chest X-rays or CT images. Laboratory data collected included routine blood tests (WBC, LYM, LYM%, neutrophil count [NEU], neutrophil percentage [NEU%], platelet count [PLT]), levels of CRP, procalcitonin (PCT) in serum, and biochemical indicators of hepatic and renal function (AST, ALT, CK, LDH, TP, ALB, UA, CREA, and BUN). In addition, according to the New Coronavirus Pneumonia Prevention and Control Program (seventh edition) [17], SARS-CoV-2 specific immunoglobulin M (IgM) and immunoglobulin G (IgG) was examined in 4 neonates' serum. The diagnosis of pneumonia in neonates was based on risk factors, clinical manifestations, radiological findings, and laboratory tests [18]. The diagnosis of COVID-19 in neonates was according to an experts′ consensus statement [19] (specific diagnostic criteria are shown in S1 Text).

## SARS-CoV-2 testing

Maternal and neonatal throat swabs and neonatal anal swabs were collected and tested for SARS-CoV-2 RNA using quantitative real time-PCR (qRT-PCR) technology and hydrolysis

probe technology (the New Coronavirus 2019 Nucleic Acid Detection Kit [Dual Fluorescence PCR] provided by Shuo Shi Biotechnology Co., Ltd, Jiangsu, China). SARS-CoV-2 specific IgM and IgG were detected using New Coronavirus 2019-nCoV IgG Antibody Detection Kit (Chemiluminescence) (Yahuilong Biological Technology Co., Ltd. Shenzhen, China) [20–22]. Sample collection, processing, and laboratory testing followed guidance from the World Health Organization [23].

## Statistical analysis

All analyses were performed using SAS 9.3 (SAS Institute, Inc, North Carolina, USA, https://www.sas.com/zh_cn/software/university-edition/download-software.html#windows). Continuous variables were described as mean (standard deviation [SD]) or median (interquartile range [IQR]). Categorical variables were represented as frequencies with proportions. All the research data were available at the ResMen Manager of Chinese Clinical Trial Registry (www.medresman.org), and the registration number is ChiCTR2000031954.

## Results

The flow chart of the study participants was shown in Fig 1. We studied 29 women with a COVID-19 infection, of which 13 were diagnosed with a SARS-CoV-2 RNA test using qRT-PCR on samples from the respiratory tract, and 16 were clinically diagnosed based on

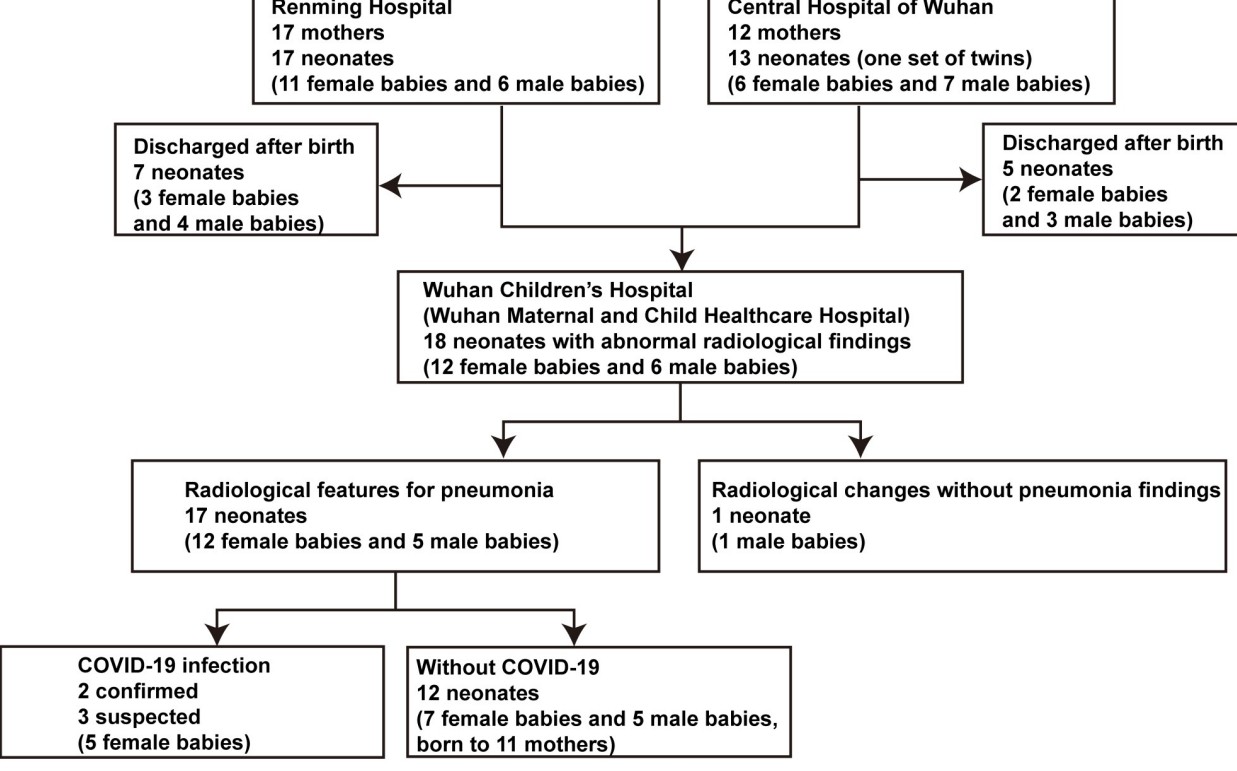

**Fig 1. Flow chart of participants inclusion.** Of the 30 neonates born (including 1 set of twins), 18 were admitted to the Children's Hospital to have further examination (5 with symptoms and 13 just for quarantine). The other 12 neonates discharged immediately did not undergo imaging screening, and follow-up showed no symptoms or abnormal findings. Among 18 neonates admitted to hospital, 17 neonates had pneumonia-like imaging changes. Five neonates were diagnosed with a COVID-19 infection, including 2 confirmed and 3 suspected. COVID-19, coronavirus disease 2019.

chest CT images, as there were not enough PCR kits available. Among 29 pregnant COVID-19-infected women, 14 experienced mild symptoms including fever (8), cough (9), shortness of breath (3), diarrhea (2), vomiting (1), whereas 15 women had no symptoms. All chest CT images showed typical signs of COVID-19 pneumonia. In the 14 women with symptoms, 9 were SARS-CoV-2 RNA positive, while in the 15 symptom-free women with abnormal chest CT images, 6 were SARS-CoV-2 RNA positive. None of the women required mechanical ventilation or were admitted to the ICU.

The distribution of the demographic characteristics, antenatal clinical conditions, and laboratory tests of women is shown in Table 1. The pregnancy-related complications, mode of delivery, postpartum conditions, and laboratory tests of the mothers are shown in Table 2. Among 29 pregnant women, 11 had complications during pregnancy, including gestational hypertensive disorder (2), gestational diabetes mellitus (3), gestational anemia (3), premature rupture of membrane (5), fetal distress (3), intrahepatic cholestasis of pregnancy (1), hepatitis (1), thrombocytopenia (1), coagulopathy (2), hypothyroidism (1), and hepatitis B virus (HBV) infection (1). Although the majority of women had a cesarean section (CS) delivery for non-medical reasons, 2 women elected to have a vaginal delivery, and both their neonates had radiological changes for pneumonia but without clinical pneumonia. There was only 1 set of twins in this study, and both neonates had a pneumonia-like lung image.

Five neonates were diagnosed as having COVID-19 infection, and all were female. The clinical features of these neonates and their mothers are presented in Tables 3 and 4. Brief histories of the 5 COVID-19-infected neonates are detailed in the supporting information (S2 Text). The date of birth was defined as Day 1. Based on the typical findings in CT images (Fig 2) and positive SARS-CoV-2 RNA test in throat swab samples, Patient 5 and Patient 9 were diagnosed as confirmed COVID-19 infection. Although SARS-CoV-2 specific serum IgM and IgG were positive in Patient 10 and Patient 18, repeated throat swab tests were all negative. Combined with the radiological features of viral pneumonia, Patient 10 and Patient 18 were diagnosed as suspected COVID-19 infection. Because of the characteristic COVID-19 pneumonia manifestations of both lungs (ground-glass opacity [GGO] in bilateral peripheral lungs, Fig 2), Patient 12 was also diagnosed as suspected COVID-19 infection.

The characteristics of the 18 hospitalized neonates are shown in Tables 4 and 5, S1 Table, Fig 2, S1 and S2 Figs. All 18 neonates had a chest X-ray or CT image within 3 days of delivery with abnormal radiological findings. The typical pneumonia-like findings were found in the 5 neonates with COVID-19 and 12 other neonates (7 female and 5 male), including diffuse or scattered patchy obscure shadows of unilateral or bilateral lungs, partial GGO, and multifocal consolidations, mostly in the lower lobes (Fig 2 and S1 Fig). The other male neonate with mild birth asphyxia showed increased lung markings and was later diagnosed with hypoxic-ischemic encephalopathy (HIE) (Patient 11).

All 18 hospitalized neonates were admitted to the NICU for care (Table 5), and laboratory tests were performed (see S1 Table for age specific results). Three neonates were born with congenital heart disease (patent ductus arteriosus [PDA] and atrial septal defect [ASD]), and they all had radiological features for pneumonia. One neonate with COVID-19 developed a fever (37.5˚C) on her third day of life, and one neonate had an occasional cough. Throat and anal swab samples of neonates were repeatedly tested to exclude or confirm COVID-19 infection, and 2 neonates tested positive in the second of 4 tests. After the New Coronavirus Pneumonia Prevention and Control Program (seventh edition) published on March 4, 2020 [17], SARS-CoV-2 specific IgM and IgG were implemented. A total of 4 neonates were tested, of which 2 were positive.

During specific or symptomatic treatment, all neonates showed improvement on chest X-ray or CT images (Tables 4 and 5; Fig 2 and S1 Fig). Follow-up chest X-ray films in 3 cases

**Table 1. Demographic characteristics, prepartum conditions and laboratory tests of mothers infected with COVID-19.**

| | All mothers | Mothers whose child had COVID-19 | Mothers whose child had abnormal radiological findings without COVID-19 | Mothers whose child discharged after birth |
|---|---|---|---|---|
| | (*n* = 29) | (*n* = 5) | (*n* = 12) | (*n* = 12) |
| | No. (%) | No. (%) | No. (%) | No. (%) |
| Age, mean (SD), year | 29.59 (3.56) | 29.40 (3.65) | 29.33 (4.01) | 29.92 (3.34) |
| BMI, mean (SD), kg/m$^2$ | 24.10 (5.54) | 21.94 (6.10) | 23.65 (5.43) | 25.45 (5.54) |
| Educational attainment | | | | |
| College or above | 17 (58.62) | 1 (20.00) | 9 (75.00) | 7 (58.33) |
| High school or lower | 12 (41.38) | 4 (80.00) | 3 (25.00) | 5 (41.67) |
| Occupation | | | | |
| Employed but not at a hospital | 11 (37.93) | 2 (40.00) | 6 (50.00) | 3 (25.00) |
| Employed specifically at a hospital | 4 (13.79) | 0 (0.00) | 2 (16.67) | 2 (16.67) |
| Unemployed | 14 (48.28) | 3 (60.00) | 4 (33.33) | 7 (58.33) |
| Parity | | | | |
| Primiparous | 24 (82.76) | 5 (100.00) | 10 (83.33) | 9 (75.00) |
| Multiparous | 5 (17.24) | 0 (0.00) | 2 (16.67) | 3 (25.00) |
| First signs and symptoms | | | | |
| Fever | 8 (27.59) | 2 (40.00) | 4 (33.33) | 2 (16.67) |
| Cough | 9 (31.03) | 2 (40.00) | 4 (33.33) | 3 (25.00) |
| Shortness of breath | 3 (10.34) | 1 (20.00) | 1 (8.33) | 1 (8.33) |
| Diarrhea | 2 (6.90) | 0 (0.00) | 1 (8.33) | 1 (8.33) |
| Vomiting | 1 (3.45) | 1 (20.00) | 0 (0.00) | 0 (0.00) |
| None | 15 (51.72) | 2 (40.00) | 6 (50.00) | 7 (58.33) |
| Throat swab | 13 (44.83) | 3 (60.00) | 8 (66.67) | 2 (16.67) |
| Chest CT | 29 (100.00) | 5 (100.00) | 12 (100.00) | 12 (100.00) |
| Prepartum laboratory tests | | | | |
| White blood cell count, median (IQR), ×10$^9$/L | 8.03 (4.50) | 8.03 (1.58) | 9.90 (5.89) | 7.30 (3.32) |
| <3.5 | 0 (0.00) | 0 (0.00) | 0 (0.00) | 0 (0.00) |
| 3.5–9.5 | 19 (65.52) | 4 (80.00) | 6 (50.00) | 9 (75.00) |
| >9.5 | 10 (34.48) | 1 (20.00) | 6 (50.00) | 3 (25.00) |
| Lymphocyte count, median (IQR), ×10$^9$/L | 1.19 (0.48) | 1.32 (0.41) | 1.17 (0.59) | 1.22 (0.35) |
| <1.1 | 9 (31.03) | 2 (40.00) | 4 (33.33) | 3 (25.00) |
| 1.1–3.2 | 20 (68.97) | 3 (60.00) | 8 (66.67) | 9 (75.00) |
| >3.2 | 0 (0.00) | 0 (0.00) | 0 (0.00) | 0 (0.00) |
| Lymphocyte percentage, median (IQR), % | 14.25 (6.37) | 16.70 (4.80) | 13.00 (3.55) | 15.05 (6.95) |
| C-reactive protein, median (IQR), mg/L | 3.50 (16.65) | 18.65 (15.12) | 12.20 (26.85) | 1.00 (3.50) |
| Aspartate transaminase, median (IQR), U/L | 15.60 (11.00) | 16.00 (24.50) | 15.30 (4.12) | 15.10 (11.75) |
| Alanine aminotransferase, median (IQR), U/L | 16.00 (13.60) | 25.00 (20.30) | 17.50 (12.68) | 14.50 (6.95) |
| Creatine kinase, median (IQR), U/L | 60.50 (48.50) | 28.00 (12.00) | 69.00 (53.00) | 69.50 (49.50) |
| Lactate dehydrogenase, median (IQR), U/L | 195.50 (53.50) | 201.00 (21.00) | 202.00 (92.00) | 178.50 (44.20) |
| Total protein, median (IQR), g/L | 59.60 (6.65) | 56.10 (2.70) | 62.20 (9.40) | 60.80 (5.67) |
| Albumin, median (IQR), g/L | 36.00 (4.65) | 36.30 (5.70) | 35.20 (4.68) | 36.20 (4.40) |

(*Continued*)

**Table 1.** (Continued)

| | All mothers | Mothers whose child had COVID-19 | Mothers whose child had abnormal radiological findings without COVID-19 | Mothers whose child discharged after birth |
|---|---|---|---|---|
| | (*n* = 29) | (*n* = 5) | (*n* = 12) | (*n* = 12) |
| | No. (%) | No. (%) | No. (%) | No. (%) |
| Uric acid, median (IQR), μmol/L | 296.00 (102.50) | 276.00 (69.00) | 295.00 (119.70) | 308.00 (109.20) |
| Creatinine, median (IQR), μmol/L | 44.00 (9.20) | 44.80 (11.10) | 44.50 (10.23) | 43.75 (6.75) |
| Urea nitrogen, median (IQR), mmol/L | 2.64 (0.65) | 2.16 (0.76) | 2.42 (0.53) | 2.70 (0.17) |

Data are presented as the mean (SD), median (IQR), or frequency (proportion). SI conversion factors: To convert aspartate transaminase, alanine aminotransferase, creatine kinase, and lactate dehydrogenase to μkat/L, multiply values by 0.0167.

BMI, body mass index; COVID-19, coronavirus disease 2019; CT, computed tomography.

**Table 2. Pregnancy-related complications, postpartum conditions, and laboratory tests of mothers infected with COVID-19.**

| | All mothers | Mothers whose child had COVID-19 | Mothers whose child had abnormal radiological findings without COVID-19 | Mothers whose child discharged after birth |
|---|---|---|---|---|
| | (*n* = 29) | (*n* = 5) | (*n* = 12) | (*n* = 12) |
| | No. (%) | No. (%) | No. (%) | No. (%) |
| Pregnancy-related complications | | | | |
| Gestational hypertensive disorder | 2 (6.90) | 0 (0.00) | 0 (0.00) | 2 (16.67) |
| Gestational diabetes mellitus | 3 (10.34) | 1 (20.00) | 0 (0.00) | 2 (16.67) |
| Gestational anemia | 3 (10.34) | 2 (40.00) | 1 (8.33) | 0 (0.00) |
| Preterm premature rupture of membranes | 5 (17.24) | 0 (0.00) | 1 (8.33) | 4 (33.33) |
| Fetal distress | 3 (10.34) | 0 (0.00) | 3 (25.00) | 0 (0.00) |
| Postpartum hemorrhage | 0 (0.00) | 0 (0.00) | 0 (0.00) | 0 (0.00) |
| Other complications | 5 (17.24) | 2 (40.00) | 1 (8.33) | 2 (16.67) |
| Mode of delivery | | | | |
| Vaginal | 2 (6.90) | 0 (0.00) | 2 (16.67) | 0 (0.00) |
| Cesarean section | 27 (93.10) | 5 (100.00) | 10 (83.33) | 12 (100.00) |
| Number of fetus | | | | |
| Singleton | 28 (96.55) | 5 (100.00) | 11 (91.67) | 12 (100.00) |
| Twin | 1 (3.45) | 0 (0.00) | 1 (8.33) | 0 (0.00) |
| Gestational age at delivery, weeks | | | | |
| 35–36 | 3 (10.34) | 0 (0.00) | 2 (16.67) | 1 (8.33) |
| 37–38 | 14 (48.28) | 3 (60.00) | 7 (58.33) | 4 (33.33) |
| 39–41 | 12 (41.38) | 2 (40.00) | 3 (25.00) | 7 (58.33) |
| White blood cell count, median (IQR), ×$10^9$/L | 8.68 (3.32) | 7.63 (3.35) | 8.47 (4.96) | 9.14 (2.28) |
| Lymphocyte count, median (IQR), ×$10^9$/L | 1.35 (0.69) | 1.31 (0.76) | 1.38 (0.57) | 1.24 (0.60) |
| Lymphocyte percentage, median (IQR), % | 15.35 (10.22) | 16.65 (11.02) | 15.65 (11.91) | 15.05 (7.39) |
| C-reactive protein, median (IQR), mg/L | 22.20 (36.39) | 52.00 (78.83) | 16.00 (32.00) | 24.70 (28.99) |

Data are presented as the median (IQR) or frequency (proportion).

COVID-19, coronavirus disease 2019.

**Table 3. Clinical features of mothers whose neonates was infected with COVID-19.**

| Patient | 5 | 9 | 10 | 12 | 18 |
|---|---|---|---|---|---|
| Symptoms | Cough | No | No | Fever, cough, vomiting | Fever, stuffy nose, shortness of breath |
| Chest CT | Patchy GGO in right lung, especially in right lower lobe | Bilateral focal GGO | Left lung infectious lesions, bilateral pleural thickening, little pleural effusion | Bilateral multiple GGO | Bilateral scattered GGO |
| Throat swab | (−) | (−) | (+) | (+) | (+) (+) |
| SARS-CoV-2 IgM, AU/mL | NA | NA | NA | NA | 279.72, Day -1; 112.66, Day 8 |
| SARS-CoV-2 IgG, AU/mL | NA | NA | NA | NA | 107.89, Day -1; 116.30, Day 8 |
| White blood cell count, ×109/L | 8.19 | 13.7 | 4.69 | 6.61 | 8.03 |
| Lymphocyte count, ×109/L | 1.49 | 1.32 | 0.78 | 1.58 | 1.08 |
| Lymphocyte percentage, % | 18.2 | 9.6 | 16.7 | 23.9 | 13.4 |
| C-reactive protein, mg/L | 20.4 | 16.9 | NA | 7 | 57 |
| Aspartate transaminase, U/L | 16 | 10 | 15.5 | 56 | 40 |
| Alanine aminotransferase, U/L | 25 | 14 | 17.7 | 54.7 | 38 |
| Creatine kinase, U/L | 28 | 56 | 35 | 23 | 17 |
| Lactate dehydrogenase, U/L | 201 | 216 | 117 | 195 | 222 |
| Total protein, g/L | 56.9 | 62.4 | 56.1 | 50.8 | 54.2 |
| Albumin, g/L | 36.3 | 38.8 | 31.5 | 26.5 | 37.2 |
| Uric acid, μmol/L | 339 | 456 | 276 | 270 | 259 |
| Creatinine, μmol/L | 51 | 45 | 44.8 | 33.9 | 28 |
| Urea nitrogen, mmol/L | NA | NA | 1.4 | 2.91 | NA |
| Treatment | Oseltamivir, Methylprednisolone, Peramivir | No | Ribavirin, Abidol | Ribavirin, Nifedipine, Clindamycin, Heparin | Oseltamivir, Methylprednisolone, Abidol, Ribavirin |
| Pregnant-related complications | No | GDM | ICP, anemia, HBV infection | Anemia | Thrombocytopenia |
| Mode of delivery | CS | CS | CS | CS | CS |
| COVID-19 diagnosis | Clinical diagnosed | Clinical diagnosed | Confirmed | Confirmed | Confirmed |

SI conversion factors: To convert aspartate transaminase, alanine aminotransferase, creatine kinase, and lactate dehydrogenase to μkat/L, multiply values by 0.0167.

COVID-19, coronavirus disease 2019; CS, cesarean section; CT, computed tomography. GDM, gestational diabetes mellitus; GGO, ground-glass opacity; HBV, hepatitis B virus; ICP, intrahepatic cholestasis of pregnancy; IgG, immunoglobulin G; IgM, immunoglobulin M; NA, not available; SARS-CoV-2, severe acute respiratory syndrome coronavirus 2.

(Patients 9, 10, 18) were performed showing increased lung markings and/or scattered obscure shadows. Before discharge, chest X-ray films in 4 cases (Patients 5, 9, 10, 12) indicated that pulmonary lesions had been completely or partially diminished, whereas the chest CT image in 1 neonate (Patient 18) had increased density in previously existing pulmonary lesions with interlobular septal thickening.

**Table 4. Clinical features of 5 neonates with COVID-19.**

| Patient | 5 | 9 | 10 | 12 | 18 |
|---|---|---|---|---|---|
| Days from mother diagnosed, days | 0 | 0 | 0 | 14 | 10 |
| Sex | Female | Female | Female | Female | Female |
| Gestational age, weeks | 40+4 | 39+1 | 38+3 | 37+1 | 37+6 |
| Birthweight, g | 3,360 | 3,570 | 2,760 | 2,940 | 3,120 |
| AGA | AGA | AGA | AGA | AGA | AGA |
| Apgar score at 1 minute | 9 | 9 | 10 | 10 | 9 |
| Apgar score at 5 minutes | 10 | 10 | 10 | 10 | 10 |
| Congenital anomaly | No | No | No | ASD | No |
| Symptoms | No | 37.5°C, Day 3 | No | No | No |
| Throat swab | (−) (+) (−) (−) | (+?) (−) (+) (−) (−) (−) | (−) (−) (−) | (−) (−) (−) | (−) (−) (−) (−) |
| Anal swab | (−) | (−) | (−) | (−) | (−) |
| Chest X-ray or CT after birth, Fig 2 | GGO | GGO | GGO | GGO | GGO |
| White blood cell count, ×109/L | 19.23 | 14.27 | 16.59 | 18.79 | 19.29 |
| Lymphocyte count, ×109/L | 2.61 | 2.19 | 2.45 | 2.95 | 4.3 |
| Lymphocyte percentage, % | 13.6 | 15.3 | 14.8 | 15.7 | 22.3 |
| C-reactive protein, mg/L | <0.75 | 11.4 | <0.75 | 1.09 | <0.79 |
| Procalcitonin, ng/mL | (−) | 2.95 | NA | NA | NA |
| SARS-CoV-2 IgM, AU/mL | NA | 0.46, Day 26 | 10.65, Day 28 | NA | 45.83, Day 1; 11.75, Day 15 |
| SARS-CoV-2 IgG, AU/mL | NA | 2.36, Day 26 | 80.46, Day 28 | NA | 140.32, Day 1; 69.94, Day 15 |
| Treatment | Symptomatic treatment | Interferon inhalation, Amoxicillin-Clavulanate potassium | Symptomatic treatment | Interferon inhalation | Interferon inhalation |
| NICU stay, days | 14 | 16 | 29 | 10 | 23 |
| Hospitalization stay, days | 15 | 16 | 29 | 16 | 23 |
| X-ray or CT before discharge, Fig 2 | No abnormal findings | No abnormal findings | No abnormal findings | Partial absorption of pneumonia | Increased densities with interlobular septal thickening |
| COVID-19 diagnosis | Confirmed | Confirmed | Suspected | Suspected | Suspected |

AGA, appropriate for gestational age; ASD, atrial septal defect; COVID-19, coronavirus disease 2019; CT, computed tomography; GGO, ground-glass opacity; IgG, immunoglobulin G; IgM, immunoglobulin M; NA, not available; NICU, neonatal intensive care unit; SARS-CoV-2, severe acute respiratory syndrome coronavirus 2.

The average NICU stay was 13 days, but 3 neonates with abnormal radiological findings without COVID-19 infection had neonatal necrotizing enterocolitis (NEC) and were therefore admitted longer. These neonates (Patient 2, 14, 15) developed NEC after birth at $37^{+6}$, $40^{+1}$, $37^{+2}$ weeks' gestation, respectively (Table 5). The first symptoms included bloody stools (Patient 2 and Patient 14, both at Day 4), repeated vomiting and bloating (Patient 15 at Day 2). Further, abdominal X-rays showed abnormalities including intermediate abdominal pneumatosis, and indistinct intestinal space (S2 Fig).

## Discussion

### Main findings

In this study, among 29 women with confirmed or clinical diagnosed COVID-19 infection, 5 neonates were diagnosed with a COVID-19 infection (2 confirmed and 3 suspected). None of these 5 neonates had contact with infected patients, except for their mothers during delivery, which suggests vertical and intrapartum transmission.

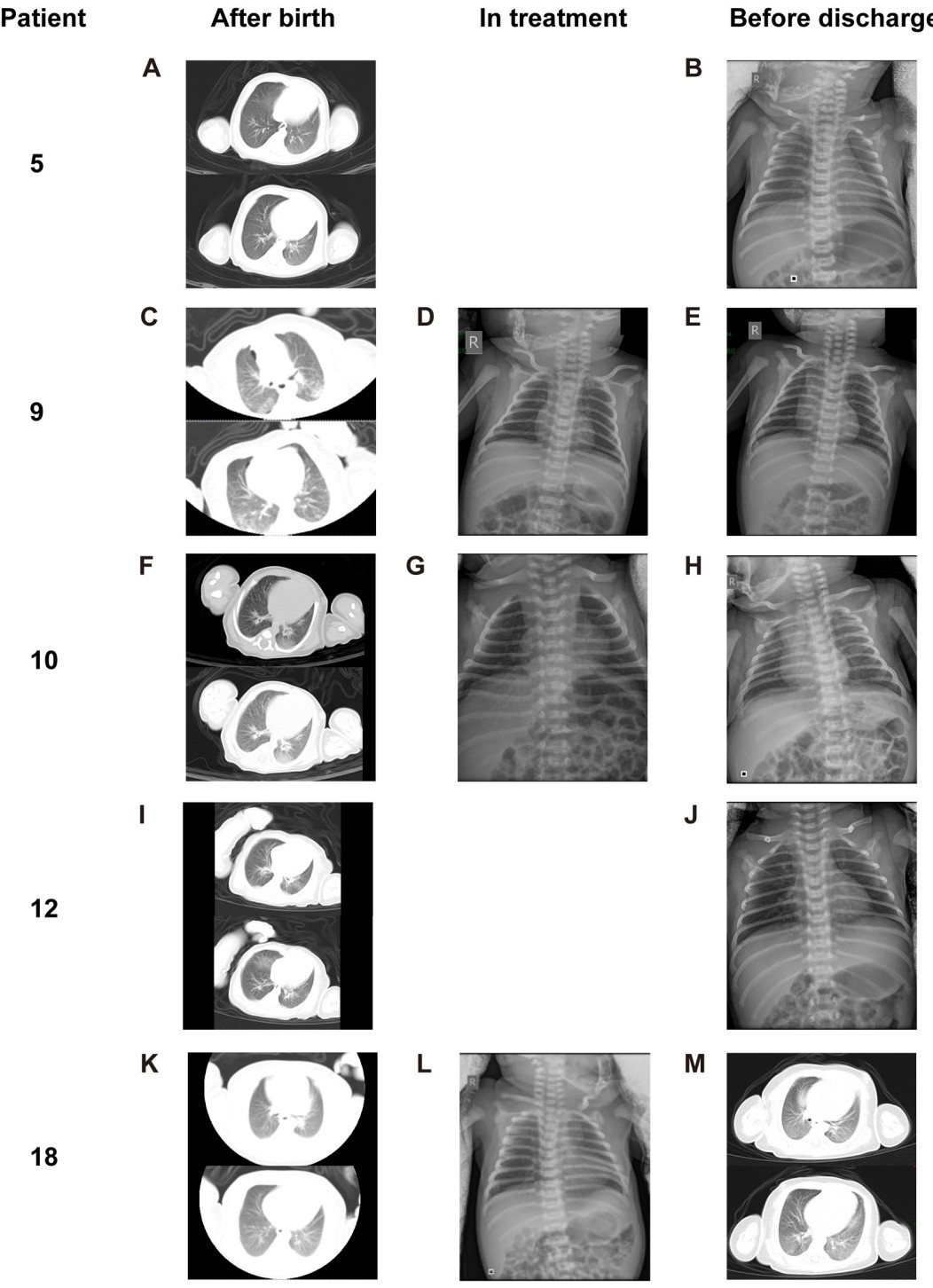

**Fig 2. Chest X-ray or CT (transverse plane) images of 5 neonates diagnosed with COVID-19 infection (2 confirmed, 3 suspected) after birth, in the treatment, and before discharge.** (A-B, Patient 5) Chest CT images (A) obtained after birth showed peripheral GGOs in the posterior basal segment of left lung lower lobe. Chest X-ray film (B) before discharge showed no abnormal findings. (C-E, Patient 9) Chest CT images (C) showed diffuse GGO with multifocal consolidations in the peripheral regions of bilateral lungs. Chest X-ray film (D) in treatment showed patchy obscure shadows in bilateral lung lower fields. Chest X-ray film (E) before discharge showed complete disappearance of pneumonia. (F-H, Patient 10) Chest CT images (F) obtained after birth showed patchy GGO in the posterior basal segment of right lower lobe and large GGO in the posterior basal segment of left lower lobe. Chest X-ray film (G) in treatment showed significant release of pneumonia. Chest X-ray film (H) before discharge showed nothing abnormal findings except mild increased lung markings. (I-J, Patient 12)

Chest CT images (I) obtained after birth showed peripheral focal consolidations in the posterior basal segments of bilateral lower lobes and multiple patchy GGO bilaterally. Chest X-ray film (J) showed partial absorption of pneumonia. (K-M, Patient 18) Chest CT images (K) obtained after birth showed reduced latency with GGO in bilateral lower lobes. Chest X-ray film (L) in treatment showed increased bilateral lung markings. Chest X-ray film (M) before discharge showed increased densities of previous existing pulmonary lesions with interlobular septal thickening. All images have been de-identified to protect patient privacy. COVID-19, coronavirus disease 2019; CT, computed tomography; GGO, ground-glass opacity.

## Results in context

Several case series have been reported focusing on the impact of COVID-19 on pregnant women and neonates [6–12]. Chen and colleagues [6] first evaluated the clinical characteristics of COVID-19 infection in 9 mother–neonate pairs. Amniotic fluid, cord blood, neonatal throat swab, and breastmilk samples were tested from 6 patients for SARS-CoV-2, and all were negative. In these case series or case reports, the majority COVID-19-infected pregnant women had mild symptoms and were delivered by CS. Further, all women had lung radiological changes, which is consistent with our results. Although in the study by Zhu and colleagues, 10 neonates experienced several clinical problems including premature labor, shortness of breath, thrombocytopenia, and even death [9]; in other reports, positive neonatal outcomes were minimal. Only 2 cases of confirmed neonatal COVID-19 infection have been reported [24]. One neonate was diagnosed at 17 days postdelivery after close contact with a COVID-19-infected mother and babysitter [25]. The other neonate was found to be infected 36 hours after a CS delivery after negative cord blood, amniotic fluid, and placenta SARS-CoV-2 RNA results [26].

In this present study, all neonates with COVID-19 infection were born via CS where both general hospitals had only 1 negative-pressure operating room. When the neonates were born through CS, they stayed unprotected in the operating room for about 30 minutes for observation before they were transferred to isolated neonatal wards. Although a guideline released by the Hubei government required immediate isolation of neonates from infected mothers [27], the 30 minutes' observation had not been noticed by obstetricians. The contaminated air in the operation room might be the source of the viral transmission, although other transmission routes cannot be excluded.

## Strengths and limitations

This is the first study to our knowledge to integrate the complete data of mothers and neonates to explore the association between maternal COVID-19 infection and neonatal COVID-19 infection or other health conditions. All women with confirmed or clinical diagnosed COVID-19 infections who gave birth at 2 of the 5 designated general hospitals between January 30 to March 10, 2020 during the most severe time of the epidemic in China were included in this study. The detailed clinical course of each hospitalized neonates was recorded and the condition of the nonhospitalized neonates was monitored by telephone interview, thus providing reliable follow-up data, which have eliminated the recall bias often found in retrospective studies. However, in the early stage of the outbreak, there was a shortage of COVID-19 test kits. As such, most neonates were only tested twice with throat and anal swabs. Only 4 neonates were tested for SARS-CoV-2 specific IgM and IgG. Ai and colleagues [28] reported chest CT had higher sensitivity in diagnosing a COVID-19 infection in comparison to qRT-PCR tests from swab samples in China. Although more than half of the pregnant women in this study were clinically diagnosed based on chest CT images due to the lack of PCR kits, we believe the diagnosis was accurate. Another limitation is the lack of tracheal aspirate to identify specific pathogens from the first 12 neonates with radiological features for pneumonia. This

**Table 5. Characteristics of hospitalized neonates without COVID-19.**

| Patient | 1 | 2 | 3 | 4 | 6 | 7 | 8 | 11 | 13 | 14 | 15 | 16 | 17 |
|---|---|---|---|---|---|---|---|---|---|---|---|---|---|
| Days from mother diagnosed, days | 7 | 7 | 1 | -2 | -2 | 0 | 0 | -2 | -1 | 1 | 0 | 0 | 0 |
| Sex | Female | Female | Female | Female | Male | Female | Male | Male | Male | Male | Female | Female | Male |
| Gestational age, weeks | 37+1 | 37+6 | 35+6 | 35+5 | 39+5 | 39+4 | 39+0 | 38+4 | 37+2 | 40+1 | 37+2 | 37+2 | 38+1 |
| Birthweight, g | 2,890 | 3,400 | 2,830 | 2,300 | 3,450 | 2,650 | 3,000 | 2,650 | 2,900 | 3,830 | 2,350 | 2,620 | 2,930 |
| AGA | AGA | AGA | AGA | AGA | AGA | SGA | AGA | SGA | AGA | AGA | SGA | AGA | AGA |
| Apgar score at 1 minute | 9 | 9 | 9 | 9 | 9 | 9 | 9 | 7 | 10 | 10 | 10 | 10 | 10 |
| Apgar score at 5 minutes | 10 | 10 | 10 | 10 | 10 | 10 | 9 | 9 | 10 | 10 | 10 | 10 | 10 |
| Congenital anomaly | No | No | No | No | No | No | PDA | No | No | No | No | No | PDA |
| Fever | No | No | No | No | No | No | No | No | No | No | No | No | No |
| Other symptoms | Occasional cough | NEC (bloody stools, Day 4) | No | No | No | No | No | Mild birth asphyxia | No | NEC (bloody stools, Day 4) | NEC (repeated vomiting and bloating, Day 2) | No | No |
| Throat swab | (−) | (−) | (−) | (−) | (−) | (−) | (−) | (−) | (−) | (−) | (−) | (−) | (−) |
| Anal swab | (−) | (−) | (−) | (−) | (−) | (−) | (−) | (−) | (−) | (−) | (−) | (−) | (−) |
| Chest X-ray or CT after birth | Patchy obscure shadow in the left lower lung field | Scattered patchy shadows in bilateral lung fields | Scattered patchy shadows in bilateral lung fields | Scattered patchy shadows in bilateral lung fields | Focal patchy consolidation is infiltrated around the left lower bronchial and paramediastinal emphysema | Scattered patchy shadows in bilateral lung fields | Focal GGO located in the medial basal segment of right lower lobe | Increased bilateral lung markings | Multifocal patchy GGOs bilaterally and focal consolidation located in the medial basal segment of right lower lobe | Pneumonia, bilateral thickened interlobular fissure | Decreasing changes of pulmonary lucency presenting as GGO in bilateral lower lobes and focal consolidation around the left lower bronchial | Multifocal consolidations located in the peripheral region of bilateral lower lobes | Diffuse GGO with multifocal consolidations located in bilateral lower lobes |
| Treatment | Sulbactam-Cefoperazone | Meropenem (for NEC) | Symptomatic treatment | Symptomatic treatment | Ceftazidime | Amoxicillin-Clavulanate potassium | Interferon inhalation | Amoxicillin-Clavulanate potassium | Interferon inhalation | Ceftazidime | Interferon inhalation, Ceftazidime, NEC treatment | Interferon inhalation, Ceftazidime | Interferon inhalation, Amoxicillin-Clavulanate potassium |
| NICU stay, days | 11 | 14 | 11 | 13 | 11 | 7 | 11 | 8 | 15 | 16 | 30 | 14 | 15 |
| Hospitalization stay, days | 11 | 17 | 11 | 28 | 14 | 7 | 14 | 26 | 15 | 16 | 30 | 14 | 15 |
| X-ray or CT before discharge | No abnormalities | Slightly increasing lung markings | Slightly increasing lung markings | Slightly increasing lung markings | No abnormalities | Slightly increasing lung markings | No abnormalities | Increased bilateral lung markings (Day 41) | No abnormalities | No abnormalities | Scattered patchy blurry shadows | No abnormalities | Pulmonary lesions are almost resolved |
| Discharge diagnosis | Hyperbilirubinemia | NEC | Hyperbilirubinemia | Premature infant | Hyperbilirubinemia | Myocardial damage | PDA | HIE | Hyperbilirubinemia | NEC | NEC | Hyperbilirubinemia | PDA |

(−), negative; (+), positive; those who tested negative for multiple times also showed (−).

AGA, appropriate for gestational age; COVID-19, coronavirus disease 2019; CT, computed tomography; GGO, ground-glass opacity; HIE, hypoxic-ischemic encephalopathy; NEC, neonatal necrotizing enterocolitis; PDA, patent ductus arteriosus; SGA, small for gestational age.

led to the uncertain etiology of the identified pneumonia. Because we found only 5 neonates with (suspected) COVID-19 infection, comparison between groups had limited statistical power.

## Research implications

Our findings suggest maternal CRP levels may be associated with neonatal lung radiological changes, as antenatal CRP levels were higher in mothers of neonates who had abnormal radiological findings (with COVID-19 or without). CRP is the most commonly researched acute-phase reactant in infection and noninfectious inflammation. There is some evidence to suggest antenatal CRP levels are associated with pregnancy-related complications [29], mental disorders in offspring [30], infant birthweight [31], and neonatal CRP levels [32]. Although CRP cannot be transmitted to the fetus through the maternal–fetal interface, maternal inflammation can affect the fetal immune state including immune cells, cytokines, and chemokines [33].

In the 5 COVID-19-infected neonates in our study (Table 4), we found their clinical manifestations were very different from those of infected adults and even children [19]. Neonates had almost no symptoms, and it was difficult to judge the changes in white blood cells and lymphocytes because of the differing number of days since birth. In terms of the SARS-CoV-2 RNA tests in throat and anal swab samples, there may have been sampling errors and false negative or false positive results. As for the detection of the SARS-CoV-2 specific IgM and IgG, only 4 cases were tested in this study due to late arrival of the testing platform, which may become a powerful method for identifying the infections in the future. When comparing all the various diagnostic examinations, the chest CT images appear to be the most important for a diagnosis and prognosis of COVID-19 among neonates. The typical radiographic findings of COVID-19 pneumonia in those 5 infants were predominately manifested as focal or diffuse GGOs located in the peripheral regions, which were distinguishable from bacterial infection but nonspecific from other virus pneumonia, such as influenza A. After treatment, all neonates showed improvement on chest X-ray or CT images. However, in 1 neonate (Patient 18), we found the densities of previous existing lung lesions increased with interlobular septal thickening, suggested that interstitial fibrosis had developed and may influence pulmonary function in the future. As previously reported [34], CT scan is more sensitive than X-ray film in detecting pulmonary lesions which are usually overlapped by the shadow of the heart or hidden behind posterior region of the diaphragms. Considering the safety of ionizing and the need of avoid wasting a medical resource, it is urgent to develop an effective standard to screen infants and children for COVID-19.

Notably, although 12 of the 13 other hospitalized neonates presented radiological features for pneumonia through X-ray or CT, only 1 neonate had an occasional cough, and none of them had any of the usual symptoms associated with a SARS-CoV-2 infection. It seems that neonates born to mothers with COVID-19 had a significantly increased risk for pneumonia-like damage within 6 days of birth. Generally, the incidence rate of neonatal pneumonia is <1% among full-term infants and approximately 10% in preterm infants [35,36]. However, the neonates in this study had no specific symptoms of pneumonia, so no microbiological test had been implemented except for SARS-CoV-2. The increased radiological features for pneumonia may be due to false negative of SARS-CoV-2 PCR test, which has a 30% false negative rate [37]. For example, Patient 12 had a typical CT feature for COVID-19 infection, but her throat and anal swabs were all negative. As such, it is possible that some of these 12 neonates had a potential SARS-CoV-2 infection and that early tracheal aspirate should be obtained for specific pathogen identification. Further, inhalation of amniotic fluid and neonatal wet lung syndrome may also cause abnormal radiological findings. Following treatment, all the lesions

shown in S1 Fig resolved, but it is unknown if any long-term effects will develop and warrants investigation.

For SARS-CoV-2 specific serum IgM and IgG, 2 neonates were positive (Patient 10 and 18) raising the concern of intrauterine transmission. Currently, SARS-CoV-2 is thought to primarily exist in the respiratory tract and to a lesser extent in the digestive system. Like other respiratory viruses, the positive rate of nucleic acid testing in blood is not high. The vertical transmission from mother-to-fetus via placenta usually requires a high replication level of virus in the blood. This may be the reason why no SARS-CoV-2 was found in the cord blood, amniotic fluid, or placenta as they were in previous studies [6,38]. When the SARS-CoV-2 specific IgM and IgG test in serum was proposed, neonates who had not been discharged were tested. IgM is the earliest synthesized and secreted antibody during human development. The fetus at the late stage of development begins to synthesize IgM, which is too large to be passed through the placenta and into the fetus. The rise of IgM is generally considered to be an indicator of intrauterine or perinatal infection. On the other hand, IgG appears to be passively transported from the mother to the fetus because it can pass through the placenta and can be synthesized by the infant after 3 months of age [39]. It is unfortunate both cases had no retained cord blood, placenta, or amniotic fluid samples so that we could test for the antibodies. Additional research is needed to further clarify vertical transmission.

In addition, 3 neonates developed NEC despite the lack of prematurity. Similar findings were found among the 5 infants born to infected women during the severe acute respiratory syndrome (SARS) outbreak [40]. One of the 5 infants born to SARS-infected women developed neonatal respiratory distress syndrome (NRDS) and jejunal perforation, and another developed NEC with ileal perforation shortly after birth. These 2 cases included extremely premature infants in which one was born at 26 weeks' gestation, and the other was born at 28 weeks. The occurrence of NEC in this study is concerning and warrants further investigation.

## Conclusions

Neonates born to COVID-19-infected women had an increased risk of COVID-19 and radiological features for pneumonia. Intrapartum and postpartum exposure to their mother may have played a role and intrauterine transmission should not be dismissed. Serum test of SARS-CoV-2 specific IgM and IgG is a valuable test to compensate for a false PCR result. The increased occurrence of necrotizing enterocolitis among neonates born to COVID-19-infected women warrants further investigation.

## Supporting information

**S1 STROBE Checklist. STROBE checklist of item that should be included in reports of cohort studies.** STROBE, Strengthening the Reporting of Observational Studies in Epidemiology.
(DOCX)

**S1 Text. Analysis plan.**
(DOCX)

**S2 Text. Brief medical histories of 5 neonates with COVID-19 infection.** COVID-19, coronavirus disease 2019.
(DOCX)

**S1 Table. Laboratory tests of all hospitalized neonates born to mother with COVID-19.**
COVID-19, coronavirus disease 2019.
(DOCX)

**S1 Fig. Chest X-ray or CT (transverse plane) images of 13 neonates with radiological change in chest but who were not diagnosed with COVID-19.** Left panel of each case was the chest X-ray or CT performed after birth (within 3 days, mostly within 24 hours). Except one showing increased bilateral lung markings (H, Patient 11), the other 12 showed the typical manifestations of pneumonia. The main findings was diffuse or scattered patchy obscure shadows of unilateral or bilateral lungs, 5 cases with partial ground-glass opacities (G, I, J, K, M, Patient 8, 13, 14, 15, 17), 5 cases with peripheral focal consolidations (E, I, K, L, M, Patient 6, 13, 15, 16, 17), and one case with paramediastinal emphysema (E, Patient 6). Right panel of each case was the chest X-ray performed before discharge (Patient 11 was performed after discharge), of which 12 neonates with pneumonia-like radiological changes showed absorption of lesions. Four cases showed slightly increasing lung markings (B, C, D, F, Patient 2, 3, 4, 7). Only one case showed scattered patchy blurry shadows (K, Patient 15). The other one still showed increased bilateral lung markings (H, Patient 11). All images have been de-identified to protect patient privacy. COVID-19, coronavirus disease 2019; CT, computed tomography. (TIF)

**S2 Fig. Abdominal X-ray images of 3 neonates with necrotizing enterocolitis.** (A, Patient 2) Abdominal X-ray images at Day 5 showed intermediate abdominal pneumatosis, indistinct intestinal space between bowels in the middle and lower abdomen, and the presence of "bubble sign". (B, Patient 14) Abdominal X-ray images at Day 5 showed abdominal pneumatosis; indistinct intestinal space between bowels in the right lower abdomen, and slightly linear and bubble-like radiolucencies. (C, Patient 15) Abdominal X-ray images at Day 20 showed mild abdominal pneumatosis, which was mainly located in the colon, and slightly indistinct intestinal space between bowels in the left lower abdomen. All images have been de-identified to protect patient privacy.
(TIF)

## Acknowledgments

Thanks to all the hospital staffs fighting in the clinical frontline.

## Author Contributions

**Conceptualization:** Yan-Ting Wu, Guo-Ping Xiong, Jing Yang, He-Feng Huang.

**Data curation:** Yan-Fen Chen, Wen Yang, Yang Chen, Han Liu, Bi-Heng Cheng.

**Formal analysis:** Chen Zhang, Andrew Kawai.

**Funding acquisition:** Yan-Ting Wu, Cheng Li.

**Investigation:** Jun Liu, Ling Jiang, Zhao-Xia Qian.

**Methodology:** Ben Willem Mol.

**Project administration:** Yan-Ting Wu, Jun Liu.

**Resources:** Yan-Fen Chen, Wen Yang, Yang Chen, Bi-Heng Cheng.

**Supervision:** Jing Yang, He-Feng Huang.

**Writing – original draft:** Jing-Jing Xu, Cheng Li, Yu Wang, Ben Willem Mol, Cindy-Lee Dennis.

**Writing – review & editing:** Yan-Ting Wu, He-Feng Huang.

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
