## [Editor Report · Decision Letter 0]

9 Apr 2020

Dear Dr Huang, 

Thank you for submitting your manuscript entitled "Neonatal outcome in 29 pregnant women with COVID-19" for consideration by PLOS Medicine.

Your manuscript has now been evaluated by the PLOS Medicine editorial staff as well as by an academic editor with relevant expertise and I am writing to let you know that we would like to send your submission out for external peer review.

Please also confirm whether this is a separate cohort from the one reported in JAMA Pediatrics recently:

https://jamanetwork.com/journals/jamapediatrics/fullarticle/2763787

Kind regards,

Thomas J McBride, PhD,

PLOS Medicine

---

## [Decision Letter · Decision Letter 1]

30 Apr 2020

Dear Dr. Huang,

Thank you very much for submitting your manuscript "Neonatal outcome in 29 pregnant women with COVID-19" (PMEDICINE-D-20-01174R1) for consideration at PLOS Medicine. 

Your paper was evaluated by a senior editor and discussed among all the editors here. It was also discussed with an academic editor with relevant expertise, and sent to four independent reviewers, including a statistical reviewer. The reviews are appended at the bottom of this email and any accompanying reviewer attachments can be seen via the link below:

[LINK]

In light of these reviews, I am afraid that we will not be able to accept the manuscript for publication in the journal in its current form, but we would like to consider a revised version that addresses the reviewers' and editors' comments. Obviously we cannot make any decision about publication until we have seen the revised manuscript and your response, and we plan to seek re-review by one or more of the reviewers. 

We expect to receive your revised manuscript by May 07 2020 11:59PM. Please email us (plosmedicine@plos.org) if you have any questions or concerns.

We look forward to receiving your revised manuscript. 

Sincerely,

Thomas McBride, PhD

Senior Editor 

PLOS Medicine

plosmedicine.org

1- Thank you for noting your intent to share the data. At this time, please provide the link to the data repository and accession numbers required for access. If you intend to include the data as a supplementary file, please include the data files with your revision and reference the supplemental file from the main text (in the Methods section).

2- Thank you for providing your STROBE statement. Please replace the page numbers with paragraph numbers per section (e.g. "Methods, paragraph 1"), since the page numbers of the final published paper may be different from the page numbers in the current manuscript.

3- If a prospective analysis plan (from your funding proposal, IRB or other ethics committee submission, study protocol, or other planning document written before analyzing the data) was used in designing the study, please include the relevant prospectively written document with your revised manuscript as a supplemental file. If no such document currently exists, please provide a supplemental file detailing the original analysis plan.

The analysis plan and your methods should clearly state (1) the specific hypotheses you intended to test, (2) the analytical methods by which you planned to test them, (3) the analyses you actually performed, and (4) when reported analyses differ from those that were planned, provide transparent explanations for differences that affect the reliability of the study's results. For example, if a reported analysis was performed based on an interesting but unanticipated pattern in the data, please be clear that the analysis was data-driven. If hypotheses that were not included in the original study design later became important to test because new evidence became available from other studies, please explain the situation, so that it is clear whether new analyses were data driven or added for another reason. Please see our guidelines for observational studies here: http://journals.plos.org/plosmedicine/s/submission-guidelines#loc-guidelines-for-specific-study-types

4- Thank you for noting Dr. Mol’s competing interests. Please clarify the last sentence “No potential conflicts of interest.” If it is meant to refer to the grant and consultancies reported by Dr. Mol, this sentence can be rephrased to say “These consultancies are not related to the current work” or similar. If it is meant to refer to the other authors, please rephrase to say “All other authors report no potential competing interests.” or similar. Either way, please do confirm whether any other authors have relevant competing interests.

5- Please revise your title according to PLOS Medicine's style. Please place the study design ("A randomized controlled trial," "A retrospective study," "A modelling study," etc.) in the subtitle (ie, after a colon). 

6- The entire manuscript would benefit from a thorough check for grammar, spelling, and clarity.

7- In the last sentence of the Abstract Methods and Findings section, please describe the main limitation(s) of the study's methodology.

8- Abstract Conclusions: Please be more specific, and address the study implications without overreaching what can be concluded from the data; the phrase "In this study, we observed ..." may be useful.

9- The Introduction could benefit with more background on what is known and not known about COVID-19 in pregnant women, infants, and other populations. Additionally, a bit more on the setting and population of the current study would be useful.

10- Please include the Ethical approval and Informed consent sections in the Methods.

11- Results lines 197-198: please present numerators and denominators for percentages.

12- Line 206: “uneventful follow-up” could be more specific.

13- Please present and organize the Discussion as follows: a short, clear summary of the article's findings; what the study adds to existing research and where and why the results may differ from previous research; strengths and limitations of the study; implications and next steps for research, clinical practice, and/or public policy; one-paragraph conclusion.

14- Please use the "Vancouver" style for reference formatting, and present reference numbers in square brackets. Please see our website for other reference guidelines https://journals.plos.org/plosmedicine/s/submission-guidelines#loc-references

Comments from the reviewers:

Reviewer #1: I confine my remarks to statistical aspects of this paper.

These were very simple but appropriately so and I recommend publication

Peter Flom

Reviewer #2: The article indeed offer a series of relatively complete data,including Maternal demographic characteristics, delivery course, symptoms, and laboratory tests. However some researchs about the outcome of neonates born to infected women have published, the innovation of the paper needs to be improved. Suggest to transfer to Letter format.

Reviewer #3: REVIEWER COMMENTS TO AUTHOR

Thank you for the opportunity to review this paper. 

There is no doubt that the COVID-19 pandemic will be of interest to PLOS Medicine's wide general audience. I believe the content of this manuscript will lead to a substantial advance in clinical management of pregnant women (and their neonates) with COVID-19. Further, the data presented provide a substantial new insight into the pathogenesis of COVID-19. I support the publication of this manuscript.

General comments, strengths and weaknesses

I believe the authors have done a good job pulling together disparate data from two hospitals with multiple clinicians involved. Whilst I think the 17% vertical/neonatal transmission is interesting, the radiologic changes of pneumonia in the non-COVID neonates are remarkable. I think the authors should make more of this unique finding. What does it mean to have 12/30 neonates from COVID mothers with radiologic pneumonia in relatively well neonates? What is the cause of the radiologic changes? What are the potential diagnoses responsible for these radiologic changes? 

Specific points:

1. What did the researchers do and find? 

a. Line 51 - while the other 12 neonates WERE discharged after birth ….

b. Line 53 - It would be easier to understand if you wrote 'Five of the 18 hospitalized neonates…'

c. I think you should also put in a comment about the NEC in the COVID negative neonates who were not preterm.

2. What do these findings mean?

a. I think the vertical transmission and radiologic features of pneumonia should be two separate points. It is interesting that 17% of neonates developed COVID-19. It is remarkable that 12/13 (92%) of the non-COVID neonates had radiologic signs of pneumonia. This should be emphasised.

3. Abstract

a. Line 71 - manifestations TO non-pregnant women

b. Lines 88-90 - this sentence does not make sense.

c. Line 91 - presented WITH radiological features…

d. Line 96 - radiological FEATURES OF pneumonia.

4. Introduction - clear

5. Methods

a. Line 136 remove WAS

b. Line 144 maternal history should be MATERNAL PARITY

6. Results

a. Line 196 - suggest AVAILABLE rather than supply

b. Line 221-222 - S1 Table is really important. I think this should be upgraded to Table 5

c. Line 242 - I think you should add after …abnormal radiological findings… (without diagnosis of COVID-19)

7. Discussion

a. Line 265-266 - 'In the present study, a detailed item might play a role' does not make sense. I suggest re-writing the sentence.

b. Line 289 - 'false negatives or false positives.' I suggest FALSE NEGATIVE OR FALSE POSITIVE RESULTS.

c. Line 291 - I suggest changing 'proposal' to THE LATE ARRIVAL OF THE TESTING PLATFORM.

d. Lines 312-314. What does the 'pneumonia-like damage' mean to the neonate in the short term and the long term? What do you think they have as their diagnoses?

8. Conclusions - clear

Reviewer #4: The paper describes infants of 29 mothers with Covid-19. Five infants had Covid-19 (2 confirmed and 3 suspected cases). Detailed description is provided with respect to infants' symptoms, laboratory measures, and x-ray/CT images. Maternal pregnancy complications and delivery are also described. I have relatively minor comments.

Comments:

1. Abstract: Line 74. A retrospective cohort study is an analytical epidemiological study that includes a comparison group (exposed vs unexposed). I suggest this study is a descriptive follow-up study.

2. Abstract: Can you, please, indicate how many hospitalized infants were hospitalized because they needed hospitalization and how many just for the quarantine. Would these infants be otherwise discharged to home if parents did not opt for the quarantine in the hospital? 

3. It would be nice to define the diagnostic criteria for Covid-19 in mothers and in infants (expert consensus) in the Introduction or Methods (the referenced literature may not be easily accessible to all). A brief description of typical signs of Covid-19 pneumonia and how it differs from other pneumonias in newborns would be also appreciated by general medical audience.

4. Results Line 193-197: there were 29 women with Covid-19, 16 were diagnosed only by chest CT. Covid-19 symptoms were present only among 14 women. It seems that chest CT was the only diagnostic sign of Covid-19 in some mothers in this study. Is it a reliable? 

5. Can you, please, describe the NEC cases in more detail in the results section? Specifically, their gestational age at birth and how many days from birth (to NEC diagnosis)? 

6. Can you please add gestational age at delivery to Table 2 (I suggest in categories, e.g., 20-33, 34-36, 37-38, 39-41, 42+ weeks)?

Minor comments:

Line 174: Please use the reference only or specify by the last name, eg "… following the paper by Nissen..."

[LINK]

---

## [Editor Report · Decision Letter 2]

27 May 2020

Dear Dr. Huang,

Thank you very much for re-submitting your manuscript "Neonatal Outcome in 29 Pregnant Women with COVID-19: A Retrospective Study" (PMEDICINE-D-20-01174R2) for review by PLOS Medicine.

I have discussed the paper with my colleagues and the academic editor. I am pleased to say that provided the remaining editorial and production issues are dealt with we are planning to accept the paper for publication in the journal.

[LINK]

In revising the manuscript for further consideration here, please ensure you address the specific points made by the editors. In your rebuttal letter you should indicate your response to the reviewers' and editors' comments and the changes you have made in the manuscript. Please submit a clean version of the paper as the main article file. A version with changes marked must also be uploaded as a marked up manuscript file.

We look forward to receiving the revised manuscript by Jun 03 2020 11:59PM. 

Sincerely,

Thomas McBride, PhD

Senior Editor 

PLOS Medicine

plosmedicine.org

Requests from Editors:

1- Thank you for providing text excerpts for the STROBE checklist items. Please either provide the full excerpt (no “…”) or describe the section and paragraph where the item appears in the main text (e.g., “Methods, paragraph 1”).

2- Thank you for providing information on how to find the dataset. However, the file (listed as a .xls) did not open on my computer. Please make sure the format is correct.

3- Please add "in Wuhan, China" to the title.

4- Please add summary demographic details for the mothers to the abstract.

5- Abstract, line 113 (of the marked up document): please edit to: “Neonates were hospitalized if they *had* signs *of COVID-19* (5 cases) or their guardians agreed *to* a hospitalized quarantine…” Please confirm whether “signs” or “symptoms” is more accurate in this sentence.

6- Please edit the Abstract Conclusions. While we appreciate providing a specific response here, the figures presented are better suited to the Abstract Methods and Findings. A more appropriate Conclusion could be: In this study, we observed that COVID-19 or radiological features of pneumonia in some, but not all, neonates born to women with COVID-19. These findings suggest that intrauterine or intrapartum transmission is possible.”

7- Thank you for providing an Author Summary. However, this should not simply restate the information in the Abstract. Please edit the “What did the researchers do and find?” and “What do these findings mean?” to provide a non-technical summary of the findings and implications. Please also limit the sections to 2-3 bullet points each.

8- Additionally, please adjust the “What did the researchers do and find” section to reflect the reviewer comments about the neonatal pneumonia findings which were not conclusive and could be a little misleading.

9- Introduction, line 178: more accurate to say “...of all women with *confirmed* COVID-19 infectons…”?

10- Introduction line 180: as this is not a randomized trial, please refrain from using causal language (eg “effect”). Instead please change to “... to assess neonatal outcomes associated with maternal COVID-19 ifection.”

11- At line 210, we suggest adapting the text to "For this retrospective study in clinical practice, the requirement for written informed consent was waived by the ethics committee.".

12- In the results section of your main text, please state the breakdown of the neonates by sex (we may have missed this in the text or tables). 

13- At line 387, please make that "had contact".

14- At line 388, please make that "... which suggests vertical and intrapartum transmission." or similar. 

15- At line 439, please avoid claiming "the first", for example, and where necessary please amend the text to "to our knowledge the first" or similar. E.g, “This is the first study to our knowledge to explore the association between maternal COVID-19 infection and neonatal COVID-19 infection or other health conditions.”

16- At lines 107 & 481, please make that "COVID-19-infected women/neonates" or similar. 

17- Discussion, line 440: please change to: “All mothers with confirmed COVID-19 infections who gave birth…”. Please also provide a more specific time frame than “around February 2020”, and specify this includes all mothers who gave birth at a hospital in China, allowing for the possibility of births that were not identified and included in this study.

18- Line 469: more accurate to say “...limited statistical power.”?

19- Line 472: “Our findings suggest…”

20- Line 481: “In *the* five COVID-19 neonates…”

21- Please incorporate your response to reviewer 2 into the second section of the Discussion, citing the most current studies on pregnant women with COVID-19 and their children, and placing your findings in the context of these other studies.

22- In the Discussion, please discuss the limitations in a bit more depth, particularly those around the radiological findings among neonates and the diagnosis of some mothers without PCR testing.

23- S1 and S2 Table seem to be missing, please upload. Page limits should not be an issue, S1 Table can be added to the main text as Table 5.

24- Please present all new results in the Results section. The section beginning at line 507 of the Discussion seems to present new information on the follow-up of several neonates that should be in the Results section.

25- Rather than using “Case X”, please refer to “patients” or “individuals”.

26- Line 359: PLOS does not allow for data that is “not shown”. Please add this information to the tables. 

27- Please remove the information on conflicts of interest, funding and data availability from the end of the main text. This information will appear in the metadata in the event of publication (via the submission form).

---

## [Editor Report · Decision Letter 3]

24 Jun 2020

Dear Prof. Huang, 

On behalf of my colleagues and the academic editor, Dr. Jenny E Myers, I am delighted to inform you that your manuscript entitled "Neonatal Outcome in 29 Pregnant Women with COVID-19: A Retrospective Study in Wuhan, China" (PMEDICINE-D-20-01174R3) has been accepted for publication in PLOS Medicine. 

PRODUCTION PROCESS

PRESS

PROFILE INFORMATION

Thank you again for submitting the manuscript to PLOS Medicine. We look forward to publishing it. 

Best wishes, 

Thomas McBride, PhD

Senior Editor 

PLOS Medicine

plosmedicine.org